# Dependency of Tamoxifen Sensitive and Resistant ER^+^ Breast Cancer Cells on Semaphorin 3C (SEMA3C) for Growth

**DOI:** 10.3390/cells12131715

**Published:** 2023-06-25

**Authors:** Satyam Bhasin, Christopher Dusek, James W. Peacock, Artem Cherkasov, Yuzhuo Wang, Martin Gleave, Christopher J. Ong

**Affiliations:** 1Vancouver Prostate Centre, Vancouver, BC V6H 3Z6, Canada; sbhasin@prostatecentre.com (S.B.); cdusek@prostatecentre.com (C.D.); jpeacock@prostatecentre.com (J.W.P.); acherkasov@prostatecentre.com (A.C.); ywang@prostatecentre.com (Y.W.); m.gleave@ubc.ca (M.G.); 2Department of Urologic Sciences, The University of British Columbia, Vancouver, BC V5Z 1M9, Canada

**Keywords:** ER^+^ breast cancer, Semaphorin 3C, plexin-B1, siRNA knockdown, Akt-MAPK signaling, apoptosis, B1SP, tamoxifen-resistance

## Abstract

Estrogen receptor positive (ER^+^) breast cancer (BCa) accounts for the highest proportion of breast cancer-related deaths. While endocrine therapy is highly effective for this subpopulation, endocrine resistance remains a major challenge and the identification of novel targets is urgently needed. Previously, we have shown that Semaphorin 3C (SEMA3C) is an autocrine growth factor that drives the growth and treatment resistance of various cancers, but its role in breast cancer progression and endocrine resistance is poorly understood. Here, we report that SEMA3C plays a role in maintaining the growth of ER^+^ BCa cells and is a novel, tractable therapeutic target for the treatment of ER^+^ BCa patients. Analyses of publicly available clinical datasets indicate that ER^+^ BCa patients express significantly higher levels of SEMA3C mRNA than other subtypes. Furthermore, SEMA3C mRNA expression was positively correlated with ESR1 mRNA expression. ER+ BCa cell lines (MCF7 and T47D) expressed higher levels of SEMA3C mRNA and protein than a normal mammary epithelial MCF10A cell line. ER siRNA knockdown was suppressed, while dose-dependent beta-estradiol treatment induced SEMA3C expression in both MCF7 and T47D cells, suggesting that SEMA3C is an ER-regulated gene. The stimulation of ER^+^ BCa cells with recombinant SEMA3C activated MAPK and AKT signaling in a dose-dependent manner. Conversely, SEMA3C silencing inhibited Estrogen Receptor (ER) expression, MAPK and AKT signaling pathways while simultaneously inducing apoptosis, as monitored by flow cytometry and Western blot analyses. SEMA3C silencing significantly inhibited the growth of ER^+^ BCa cells, implicating a growth dependency of ER^+^ BCa cells on SEMA3C. Moreover, the analysis of tamoxifen resistant (TamR) cell models (TamC3 and TamR3) showed that SEMA3C levels remain high despite treatment with tamoxifen. Tamoxifen-resistant cells remained dependent on SEMA3C for growth and survival. Treatment with B1SP Fc fusion protein, a SEMA3C pathway inhibitor, attenuated SEMA3C-induced signaling and growth across a panel of tamoxifen sensitive and resistant ER^+^ breast cancer cells. Furthermore, SEMA3C silencing and B1SP treatment were associated with decreased EGFR signaling in TamR cells. Here, our study implicates SEMA3C in a functional role in ER^+^ breast cancer signaling and growth that suggests ER^+^ BCa patients may benefit from SEMA3C-targeted therapy.

## 1. Introduction

Hormone receptor positive (ER^+^/PR^+^/HER2^−^) breast cancer (BCa), commonly known as estrogen receptor positive (ER^+^) breast cancer, represents approximately 65–70% of new breast cancer cases diagnosed each year and accounts for the highest number of breast cancer related deaths. Of all breast cancers, 80% have positive ER status [1,2]. ER^+^ BCa cells are activated by the ER, which is a ligand-dependent transcription factor that is believed to regulate cancer growth by inducing the expression of downstream target genes involved in cell growth, survival, and proliferation [3,4]. However, the identity of ER-regulated genes that drive cancer growth and survival is poorly understood.

The ER pathway is the primary driver of ER^+^ subtypes, and as such endocrine treatments are the first-line therapy for ER^+^ BCa patients. These treatments include selective ER modulators (e.g., Tamoxifen), selective ER degraders (e.g., Fulvestrant), and aromatase inhibitors (e.g., Letrozole) [5]. While most patients initially respond to endocrine therapy, the clinical response is transient, as approximately 40% of the patients experience treatment resistance to ER-targeting therapies [6]. Although many mechanisms for achieving endocrine resistance have been described, most endocrine resistant BCa cells remain dependent on the ER pathway [7,8]. This suggests that critical ER-regulated genes are necessary for driving cancer growth and survival; therefore, the identification of new ER-regulated targets that control growth and survival is urgently needed.

Semaphorins are a group of signaling proteins with an emerging role in cancer. Recent research indicates that Semaphorin 3C (SEMA3C), a member of the semaphorin family, significantly contributes to cancer progression by promoting tumor growth, invasion, and metastasis in a variety of tumors including prostate, breast, lung, colorectal, ovarian, pancreatic and glioblastoma cancers [9,10,11]. We have previously shown that SEMA3C is an autocrine growth factor that drives cancer growth and treatment resistance through the transactivation of multiple receptor tyrosine kinases (RTK) such as HER2, Epidermal Growth Factor Receptor (EGFR), and hepatocyte growth factor receptor (MET) in a cognate ligand independent manner [10,12]. 

In breast cancer, SEMA3C silencing has been shown to inhibit the cell invasion of MCF7 and MDA-MB-231 cells [13]. Furthermore, tissue microarray analyses of 343 invasive ductal breast carcinomas revealed that SEMA3C was expressed in 90% of the cases and was positively correlated with tumor grade (*p* < 0.001) [14]. However, the mechanism by which SEMA3C drives breast cancer progression and endocrine resistance is poorly understood. The aim of this study is to further characterize the role of SEMA3C in driving growth and survival in tamoxifen-sensitive and -resistant ER^+^ breast cancer cells.

Here, we show that SEMA3C is significantly upregulated in ER^+^ breast cancer patients and cell lines, and its expression is modulated by the estrogen receptor. Importantly, we demonstrate that silencing SEMA3C inhibits the MAPK and AKT signaling pathways, promotes apoptosis, and suppresses the growth of both tamoxifen-sensitive and resistant ER^+^ breast cancer cells. We further demonstrate the druggable potential of SEMA3C by using the B1SP Fc fusion protein, a SEMA3C pathway inhibitor that has displayed inhibitory effects on the growth and signaling of ER^+^ breast cancer cells comparable with those observed during SEMA3C silencing. These findings underscore the emerging role of SEMA3C in promoting ER^+^ breast cancer growth and survival and suggest that targeting SEMA3C may represent a potential therapeutic strategy for ER^+^ breast cancer patients. 

## 2. Materials and Methods

### 2.1. Cell Lines

MCF10A and T47D were purchased from the American Type Culture Collection (ATCC, Manassas, VA, USA). MCF7 (authenticated with IDEXX Bioanalytics Case #16382-2019), TamR3 and TamC3 (authenticated with AmpFLSTR Identifiler PCR Amplification kit (Applied Biosystems, Foster City, CA, USA) according to the manufacturer’s protocol) were kindly provided by Dr A. Cherkasov (Vancouver Prostate Centre). MCF10A were maintained in growth medium as recommended by the American Type Culture Collection (ATCC, Manassas, VA, USA). MCF7 and T47D cell lines were maintained as recommended in [15]. TamC3 were maintained lacking tamoxifen, but containing the diluent (0.1% ethanol) in phenol-red-free RPMI containing 10% charcoal-stripped fetal bovine serum (CSS), and 1X Insulin-Transferrin-Selenium and TamR3 were maintained in tamoxifen (1 uM in ethanol) in phenol-red-free RPMI containing 10% charcoal-stripped fetal bovine serum and 1X Insulin-Transferrin-Selenium. All cell lines were mycoplasma free. Experiments were carried out on cells grown in their respective growth medium, unless otherwise specified.

### 2.2. Antibodies, Chemicals and Reagents

Antibodies to the following were used at 1:1000 dilution unless specified: p-EGFR (Y1068), p42/44 pMAPK(T202/Y204), p42/44 MAPK(L34F12, 1:2000), Akt (9272), p-Akt (Ser473, D9E), cleaved PARP (Asp214), PARP (9542), Estrogen Receptor alpha (D8H8) cleaved caspase-3 (Asp 175, 5A1E, 9664), caspase-3 (9662), cleaved caspase-8 (Asp 374, E6H8S,), caspase-8 (1C12, 9746) (Cell Signaling Inc., Beverly, MA, USA) SEMA3C (N-20, 1:2000); Plexin B1 (H300, 1:500), t-EGFR (C-2, 1:1000) (Santa Cruz Biotechnology, Santa Cruz, CA), Actin (A2066, 1:5000), Vinculin (1:5000) (Roche Diagnostics, Laval, PQ, Canada), LICOR secondary antibodies- anti Rabbit IR680 and IR800, anti mouse IR680 and IR800 (Rockland, Gilbertsville, PA, USA) and anti Goat680 (Pierce Biotechnology, Rockford, IL, USA). Antibodies were used according to manufacturer’s protocols. The following chemicals were used: Tamoxifen (68392-35-8; Sigma-Aldrich, Oakville, ON, Canada), Fulvestrant (I4409-25MG; MiliporeSigma, Oakville, ON, Canada), Estradiol (E2758; Sigma-Aldrich, Oakville, ON, Canada), Charcoal Strip Serum (12676029; Gibco, Grand Island, NY, USA), RNase A (12091021; Invitrogen, Carlsbad, CA, USA), Propidium Iodide (P3566; Invitrogen, Carlsbad, CA, USA) and Insulin-transferrin-selenium (41400045; Gibco, Grand Island, NY, USA).

### 2.3. External Datasets Utilized in This Study

We utilized three publicly available databases (Table 1): (1) Cbioportal [16,17], (2) GEODATSET [18] and (3) GEPIA2 [19]. 

### 2.4. RNA Extraction and qPCR

Total RNA was extracted from cultured cells using the PureLink RNA kit (Invitrogen). Total RNA (1.5–2ug) was reverse-transcribed using a High-Capacity cDNA Reverse Transcription Kit (Applied Biosystem). Real-time monitoring of PCR amplification of cDNA was performed using SEMA3C (Hs00170762_mL) and GAPDH (Hs03(9)29097_gl (Applied Biosystems) on a ABI PRISM 7900 HT Sequence Detection System (Applied Biosystems) with SYBR gene expression Master Mix (Applied Biosystems, Burlington, ON, Canada). Target gene expression was normalized to GAPDH levels in respective samples as an internal standard.

### 2.5. Immunoblotting

Cells were lysed with 100 uL lysis buffer (50mM Tris pH 7.5, 150 mM NaCl, 1% NP-40, 10 mM NaF, 10% Glycerol) containing Complete and PhosStop phosphatase inhibitors (Roche, Mississauga, ON, Canada). Whole cell lysates were centrifuged at 14,000 rpm for 15 min at 4 °C. The protein concentration in cleared whole cell lysates was identified using BCA kit (Thermo Scientific, Rockville, IL, USA). Protein lysates (30–50 ug) were analyzed by SDS-PAGE (6–10%) and transferred by Western Blot to nitrocellulose membranes (Bio-Rad, Hercules, CA, USA). Membranes were blocked in PBS containing 5% BSA or TBS Odyssey buffer (Licor, Lincoln, NE, USA) if probing with phospho-antibodies. The primary antibody dilution was conducted as per the manufacturer’s protocol. Membranes were washed 3x with TBST followed by appropriate IRdye-conjugated secondary antibodies for 1h. Blots were washed again 3X for 5 min with TBST. Detection was via image analysis using a LICOR infrared imager and Image Studio Lite version 3.1 software. 

### 2.6. Cell Stimulation 

MCF7 and T47D were seeded at 1 × 10^5^ cells/well in a six-well plate in RPMI medium containing 10% FBS for two days. After two days, the medium was replaced with RPMI phenol-red-free and serum starved for 24h. MCF10A cells were treated similarly under their respective recommended media, and for serum starvation we utilized MEBM without phenol red for 5 h. The medium was changed again just prior to stimulation to remove endogenous-secreted SEMA3C. For SEMA3C stimulations: MCF7, T47D and MCF10A were treated with SEMA3C for 10 min at the indicated concentrations followed by a wash with PBS. For B1SP, MCF7 and T47D, cells were treated with B1SP for 90 min, and following B1SP treatment cells were treated with SEMA3C for 10 min without removing media followed by a wash with PBS. To study the effect of tamoxifen on SEMA3C expression in tamoxifen-resistant cells, TamC3 and TamR3 were seeded at 5 × 10^5^ cells/mL in a 10 cm dish in their growth media conditions except with 0.1% ethanol or 1 uM tamoxifen, respectively, for three days in order to see the effect. After three days, these cells were seeded into a 6-well plate at 1 × 10^5^ cells/well and were treated with either Ethanol (95%) or tamoxifen (0.1 uM) for 48 hrs. All of these treatments were carried out at 37 °C in 5% CO_2_.

### 2.7. Plexin B1 and SEMA3C Silencing

siRNA-targeting Plexin-B1 and SEMA3C were validated: siPlexin-B1-1 (Hs_PLXNB1_6, Qiagen, Montreal, PQ)- CCGGGUGGAAUUUAUCCUUGAUU, siSEMA3C-1 (Ambion, Life Technologies Corporation, Carlsbad, CA)- CACCAUCCUUUAGACUACATT. MCF7 and T47D cells were reverse transfected with either scramble siRNA (siScr) (AUCAAACUGUUGUCAGCGCUGUU), siPlexin-B1-1 (25 nM) or siSEMA3C-1 (25 nM) using RNAiMax (Invitrogen, Mississauga, ON, Canada) as described by the manufacturer. For immunoblotting, transfected cells were incubated for 72 h then washed with PBS and then lysed. Plexin-B1 (45 ug) and SEMA3C (40 ug) levels were analyzed with SDS-PAGE. For cell growth assays, cells were reverse transfected in a 96-well flat bottom plate and proliferation was monitored over 4 days (siPlexin-B1) and 5 days (siSEMA3C) with readings taken at every alternate day starting from Day 0.

### 2.8. Cell Growth Assay

The cell growths of MCF10A (2000 cells/well), MCF7 (1500 cells/well) and T47D (1500 cells/well), TamC3 (1500 cells/well) and TamR3 (2200 cells/well) were assessed in 96-well plates using Presto Blue Viability Reagent (Life Technologies Corporation, Carlsbad, CA, USA) according to the manufacturer’s protocol. Fluorescence was measured at excitation 535 nm and emission 612 nm using a Tecan F500 plate reader (Tecan, Grodig, Austria, GmbH). To detect cell growth for (1) siRNA silencing, we conducted reverse-transfection within a 96-well plate and analyzed data at different time points. (2) For showcasing the dependence of cells on SEMA3C, we used Plexin-B1 complex for growth, and exogenous SEMA3C (1 uM) was added to wells in presence of charcoal strip serum. (3) For IC_50_, cell growth inhibition was by B1SP and during synergy assay we allowed cells to attach to the wells of the 96-well plate overnight and then administered B1SP, Fulvestrant or tamoxifen the next day at the indicated concentrations in respective experiments (see results section).

### 2.9. Sub-G0/G1 DNA Content Assay

The flow cytometry analysis of propidium iodide-stained nuclei was performed as shown previously [26]. MCF7 and T47D cells were reverse-transfected with SEMA3C siRNA (20 nM) in 6-well plates, allowed to attach overnight in RPMI containing 2% FBS, and were forward-transfected again the next day. Then, 48 h later, cells were trypsinized, washed in 1 mL PBS and then fixed in 1 mL 80% ethanol overnight at 4 °C. Fixed cells were incubated in DNase-free RNase A reagent (50 ug/mL) and propidium iodide (50 ug/mL) at 4 °C for 1 h. The stained cells were analyzed for relative DNA content on a Becton Dickesnon FACSCanto II flow cytometer (BD Biosciences, Mississauga, ON, Canada). 

### 2.10. Production of SEMA3C: FL and B1SP

For SEMA3C: FL, cells (40 million) were seeded in Cellstack (2-stack) culture chambers (Corning Incorporated, Tewksbury, MA, USA) in a volume of 200 mL DMEM, supplemented with 5% FBS (Life technologies, Gaithersburg, MD, USA), heparin (25 μg/mL) and blasticidin (5 μg/mL). Conditioned media (CM) were collected after 48 h and then replenished with the same medium containing 2% FBS. The CM were collected every 48 h and replenished. Harvested CM were then centrifuged at 2000 RPM for 5 min and then filtered (0.22 μm). Filtrates were purified over HisTrap excel (GE Healthcare, Uppsala, Sweden). Recombinant SEMA3C-FL was then buffer exchanged and concentrated on an Amicon Ultra-15 centrifugal filter device (MWCO, 10 kDa), to 2.0 mg/mL (MiliporeSigma, Oakville, ON, Canada). The protein sample was then desalted on a PD midiTrap G-25 column, as directed by the manufacturer (GE Healthcare, Buckinghamshire, UK), and then filter sterilized using a 0.22 μm filter. B1SP was produced from CHO-S B1SP Cells (2 × 10^8^), which were seeded and cultured in a 5 L Wave Bag (GE Healthcare, Uppsala, Sweden) using 75% Freestyle CHO expression medium/25% Optimem supplemented with 8 mM glutamine and penicillin/streptomycin (5.0 ml/L) (Life technologies, Gaithersburg, MD, USA) at 37 °C and 8% CO_2_. CM were harvested three times over a period of 4 weeks and B1SP was purified using the specific purification process described previously [12].

### 2.11. Statistical Analysis and Software Used

The clinical data and in vitro data were assessed using Student’s *t*-test and ANOVA. Data were expressed as the mean ± standard error of the mean (SEM). Comparisons between two means were performed using a Student’s t-test. Comparisons among multiple means were performed with a one-way ANOVA followed by Fisher’ protected least significant difference test (StatView 512, Brain Power, Inc., Calabasas, CA, USA). GraphPad Prism software (Version No. 8.4.3) was used to calculate the statistical significance and generate graphs. ChIP-Atlas was used as a source for CHIP-Seq and DNA-seq studies, and the IGV (Integrative Genomics Viewer) browser (Version 2.14.0) was used to analyze the files. Densitometry analysis of SEMA3C expression and phosphorylation of MAPK and Akt under B1SP treatment were performed using ImageJ 1.53 e (Wayne Rasband and contributors, National Institutes of Health, USA) based on two independent experiments. The Synergy Finder Online tool was used to calculate synergy scores. The thresholds of statistical significance were set at * *p* < 0.05, ** *p* < 0.01, *** *p* < 0.001, and **** *p* < 0.0001.

## 3. Results

### 3.1. SEMA3C Is Significantly Upregulated in ER^+^ Breast Cancer, and Its Expression Is Positively Correlated with the Expression of Estrogen Receptor (ER)

To assess the clinical relevance of SEMA3C in ER^+^ breast cancer, we analyzed publicly available clinical datasets to investigate the association between SEMA3C expression and patient characteristics, as well as the correlation of SEMA3C with known breast-cancer-associated genes. SEMA3C is a protein that has been implicated in various aspects of cancer development and progression and is overexpressed in several types of cancer, including breast, prostate, and lung cancer. From the analysis of normalized RNA expression data from the TCGA datasets, we found that breast tumors exhibited the highest levels of SEMA3C expression (proteinatlas.org) compared to other cancer types, and by comparing its expression in breast normal vs. breast tumors using GEPIA2, we confirmed higher mRNA expression in breast tumors than matched normal TCGA and GTex samples (*p* < 0.05) [19,27] (Appendix A).

Next, we investigated SEMA3C mRNA expression in 1764 breast cancer patients using clinical data from a METABRIC study on cbioportal (see Methods for details on the sample grouping in each dataset) [20,21]. We found SEMA3C’s mRNA expression to be the highest in ER^+/^HER2^−^ breast cancer patients when compared to all other subtypes, including HER2^+^ and Triple-Negative breast cancer (*p* < 0.0001) (Figure 1A). Furthermore, by segregating the patients based on their ER status (positive or negative) we observed that SEMA3C mRNA expression was significantly higher in ER-positive tumor samples (TCGA, Firehose Legacy), providing further confirmation of this positive association [22] (Figure 1B). 

Gene co-expression studies can shed light on the mechanisms underlying breast cancer development and progression. To define the co-expression relationship between SEMA3C and the estrogen receptor, we used mRNA expression data of SEMA3C and ESR1 from a TCGA study (Firehose Legacy-Breast Invasive Carcinoma) with 1100 samples [22]. We performed gene correlation analysis (Pearson correlation) and found a positive correlation between SEMA3C and ER-alpha mRNA expression levels (*p* < 0.0001) (Figure 1C). Interestingly, selecting ER^+^/HER2^−^ patients within this study revealed similar significant positive correlations (*p* < 0.0001) between SEMA3C and ESR1 mRNA expression (Figure 1C). Taken together, these findings suggest a potential functional relationship between SEMA3C and ER, warranting further investigations to elucidate the functional relationship between SEMA3C and ER in promoting tumorigenesis. 

### 3.2. SEMA3C Is an ER-Induced Gene That Is Highly Expressed in ER-Positive Breast Cancer Cell Lines

To investigate the potential functional or regulatory relationship between SEMA3C and ESR1, we performed a data mining analysis of GEO datasets (GSE11324, GSE27473) generated by Carroll et al. [24] and Al Saleh et al. [25], studying genome-wide effects of the activation or depletion of the ER pathway. Our analysis from these studies revealed that SEMA3C mRNA was robustly induced by estradiol in a time-dependent manner in MCF7 cells (Appendix A). Moreover, SEMA3C mRNA expression was significantly suppressed upon ER silencing, indicating that SEMA3C is an ER-regulated gene (Appendix A). Importantly, in the Cheng et al. study, where they studied upregulated genes post estrogen activation by utilizing the GSE11324 dataset, we identified SEMA3C among the top 100 genes induced by estradiol in that study (*p* < 0.0001) [28] (Figure 2A), further supporting the notion that SEMA3C expression is tightly linked to ER signaling in breast cancer. 

To further explore the molecular mechanisms underlying SEMA3C’s role in breast cancer, we conducted in vitro cell-based experiments. First, we investigated the levels of SEMA3C protein in a panel of breast cancer cell lines. Consistent with our clinical observations, we found that SEMA3C protein expression is significantly elevated in ER^+^ breast cancer cell lines, MCF7 and T47D, as compared to MCF10A, which are immortalized non-malignant breast epithelial cells that possess many characteristics of normal breast cells (Figure 2B). Our observation of higher SEMA3C expression in breast cancer cell lines corroborates the clinical observation made earlier, in which the transcript levels of SEMA3C were higher in breast cancer tissues as compared to adjacent normal mammary tissues [13]. Next, to confirm whether SEMA3C is an ER-regulated gene, we treated MCF7 and T47D cells (previously grown in charcoal stripped serum for 4 days) with estradiol in a dose-dependent manner and monitored SEMA3C mRNA and protein levels. A significant dose-dependent (0.1–1 nM) increase in SEMA3C mRNA and protein was observed after an overnight treatment with estradiol (Figure 2C and Appendix A). In addition, after identifying 1 nM as a sufficient dose of estradiol to induce SEMA3C expression, we conducted a similar study to Carroll et al., but at the protein level, to observe the gradual increase in SEMA3C expression in a time-dependent manner [22] (Figure 2C). To investigate the role of ER in regulating SEMA3C expression, we performed siRNA knockdown experiments targeting ER in MCF7 and T47D cells. Our results showed that SEMA3C protein expression levels were decreased upon ER siRNA knockdown (Figure 2C). These findings suggest that ER plays a critical role in regulating SEMA3C expression in ER^+^ breast cancer cells, further supporting the notion that SEMA3C is an ER-regulated gene.

To gain a better understanding of how ER might be regulating SEMA3C expression, we analyzed ER-ChIP Seq studies conducted on MCF7 (SRX4451213, SRX4367193, SRX4451208) and T47D (SRX5702620, SRX1961156, SRX1161199) from the ChIP-Atlas database [29,30]. Based on three independent studies for each cell line, we identified significant ER binding peaks upstream or on the SEMA3C locus (Upstream/Promoter, Intron 1, Intron 2) suggesting ER interaction on the SEMA3C locus (Figure 2D). With the help of DNA-seq studies from the ChIP-Atlas, we were able to show chromatin accessibility in these regulatory regions to allow the potential binding of ER-alpha (SRX193608, SRX193608) (Figure 2D). These ER-binding regions on the SEMA3C locus were consistent with studies conducted by Tam et al. [31]. Furthermore, these regions have also been identified as cis-Regulatory Elements (cCREs) (EH38E2566990, EH38E2566989, EH38E3782829, EH38E2566941) [32]. With help of JASPAR [33], we were able to identify multiple ER binding motifs (MA0112.2 and MA0112.3) in SEMA3C regulatory regions (Upstream/Promoter, Intron 1, Intron 2) which overlapped with the ER ChIP seq peaks observed earlier (Figure 2D and Appendix A). 

### 3.3. SEMA3C Activates RTK Signaling Pathways in Breast Cancer Cells via Plexin-B1

To investigate SEMA3C’s potential role as a driver of breast cancer, we needed to induce SEMA3C signaling via the stimulation of ER^+^ breast cancer cells and examine whether SEMA3C can activate RTK signaling pathways that are commonly observed in breast cancer. Our previous findings indicate that SEMA3C functions as a secreted soluble autocrine growth factor that drives cancer growth by transactivating multiple RTKs such as EGFR, HER2 and MET via Plexin B1 [12]. To determine whether SEMA3C can activate these RTKs, we treated MCF10A cells with recombinant SEMA3C (0–1 uM). Our results showed that MCF10A cells expressed EGFR and MET but not HER2 (Appendix A). Additionally, SEMA3C treatment induced the phosphorylation of MAPK and AKT, as depicted in (Figure 3A).

In ER^+^ breast cancer patients, the Akt and MAPK signaling pathways are known to be constitutively active due to downstream effects of ER [34,35]. To investigate whether SEMA3C activates these pathways in ER^+^ breast cancer cells, we treated MCF7 and T47D cells with recombinant SEMA3C in a dose-dependent manner. Our results indicated that SEMA3C treatment activated MAPK (Thr202/Tyr204) and Akt (Ser473) proteins in a dose-dependent manner (Figure 3B). 

Based on our previous report, which suggested that SEMA3C transactivates multiple RTKs through its interaction with Plexin B1, we conducted plexin B1 silencing experiments to investigate whether SEMA3C drives signaling through Plexin B1 in ER^+^ breast cancer. We confirmed that both MCF7 and T47D cells express high levels of Plexin B1 (Figure 3C), and established that Plexin-B1 serves as the receptor for SEMA3C in ER^+^ breast cancer cells. The siRNA knockdown (20 nM) of Plexin-B1 attenuated cell signaling and growth, even in the presence of recombinant SEMA3C (Figure 3C,D). These findings provide further evidence that SEMA3C is a potential therapeutic target in the treatment of breast cancer.

### 3.4. ER^+^ Breast Cancer Cell Lines Show Dependency on SEMA3C for Cell Growth, and Its Depletion Leads to the Induction of Pro Apoptotic Proteins

After exploring cell signaling pathways activated by SEMA3C, we wanted to understand whether ER^+^ breast cancer cells were dependent on SEMA3C-induced signaling. Therefore, we aimed to investigate whether SEMA3C silencing could impact tumor cell growth. To this end, we conducted a siRNA knockdown of SEMA3C (25 nM) on MCF7 and T47D cells, and 72 h post-treatment we analyzed cell signaling pathways known to be upregulated in ER^+^ breast cancer via immunoblotting. We observed a decrease in phospho-Akt and phospho-MAPK signals (Figure 4A), and saw a significant impact in cell growth on both of the cell models, even in the presence of serum, within five days (Figure 4B). 

As already known, the ER pathway is the main driver of tumor progression in ER^+^ breast cancer and cells. Utilizing this knowledge, we wanted to study whether the activation of the ER pathway would overcome the effect of SEMA3C silencing on growth, as observed previously. We performed reverse siRNA knockdown of SEMA3C and Scr in respective wells in a 96-well plate. The following day, we treated half of the siRNA-treated wells with 0.1 nM estradiol to activate the ER pathway, and the other half with 95% EtOH as a control in presence of CSS. Within a few days, we started to observe a similar growth inhibition pattern as seen in Figure 4B, irrespective of whether they were treated with estradiol or EtOH. Notably, the treatment with estradiol was unable to reverse the growth inhibition caused by SEMA3C knockdown (Figure 4C). This observation suggested that SEMA3C signaling might have a positive feedback effect on the ER pathway. To confirm this, we examined ER protein expression in MCF7 post-SEMA3C siRNA knockdown. As seen in Figure 4C, we confirmed that SEMA3C can affect ER expression, suggesting cross-talk between SEMA3C and the ER signaling pathway.

Next, we wanted to elucidate the mechanism underlying the impact of SEMA3C knockdown on cell growth. We examined whether this effect was, in part, due to induction of cell apoptosis. By conducting SEMA3C silencing (25 nM) in a six-well plate and collecting conditioned media, along with whole cell protein lysate, we stained the cells with propidium iodide and observed a higher percentage of cells in the sub G0/G1 fraction, which was indicative of apoptotic cell death (Figure 4D). Through immunoblotting, we were able to confirm at the protein level an increase in cleaved caspase-8 and PARP (Figure 4E) in SEMA3C knockdown cells, in addition to cleaved caspase-3 detection only in T47D, but not in MCF7, which are known to be caspase-3 deficient (Appendix A). These findings suggest that SEMA3C knockdown induces apoptosis in MCF7 and T47D cells, which may contribute to the growth inhibition observed earlier.

### 3.5. SEMA3C Pathway Inhibitor Showcases a Significant Impact on Endocrine Therapy Naive Cells Growth and Exhibits Potential Synergistic Behavior with Tamoxifen

Our study indicates that SEMA3C is a potential therapeutic target in ER^+^ breast cancer, as it drives the activation of RTK signaling pathways, and SEMA3C knockdown significantly impacts cell growth. To assess its drugability potential within ER^+^ breast cancer, we conducted proof-of-concept testing using B1SP, a novel SEMA3C pathway inhibitor, to lay the groundwork for the development of SEMA3C-targeted therapies in ER^+^ breast cancer.

As shown above, we observed that SEMA3C is involved in the activation of MAPK and AKT signaling pathways and that SEMA3C siRNA treatment has an impact on cell growth, hinting that this growth factor might have a therapeutic potential. Currently, there are no known SEMA3C-targeted therapies in the clinic. We wanted to conduct a proof-of-concept analysis confirming SEMA3C as a druggable target. Therefore, we utilized our in-house-developed therapeutic protein inhibitor of SEMA3C signaling, which was engineered as a Plexin B1: Fc fusion decoy protein to functionally disrupt SEMA3C-induced RTK activation, and is called B1SP. Its role has been validated in prostate cancer both in vitro and in vivo [12]. To determine whether B1SP can be a potential therapeutic for ER^+^ breast cancer, we tested the effect on normal mammary epithelial cells, MCF10A, and ER^+^ breast cancer cells MCF7 and T47D (Appendix A). B1SP inhibited the SEMA3C-induced activation of oncogenic signaling pathways (Akt and MAPK) in both MCF7 and T47D cells in a dose-dependent manner (Figure 5A), and then induced significant (*p* < 0.01) growth inhibitory effects on MCF7 and T47D cells within 72 h (Figure 5B). We further wanted to identify whether B1SP in combination with currently approved first- and second-line endocrine therapy (Tamoxifen and Fulvestrant) had any synergistic or antagonistic effects, by studying the effect on cellular growth in endocrine therapy naïve MCF7 and T47D cells. It is critical to assess whether B1SP can improve efficacy of these approved drugs and potentially prolong endocrine resistance. For this experiment, we created a matrix with a range of different treatment concentrations of drugs (B1SP = 0–1.6 uM; Tamoxifen = 0–1.6 uM; Fulvestrant = 0–0.5 uM) with specific culture conditions for the two combinations (B1SP–tamoxifen and B1SP–fulvestrant), and the cells were analyzed 96 h post-treatment. For the experiment, the expected drug combination responses were calculated based on the ZIP reference model, using SynergyFinder to identify relationships between B1SP and the endocrine therapy drugs. For the B1SP–tamoxifen combination, potential synergy was observed based on ZIP synergy score (≥10 likely synergistic or additive ≥ −10, ≤10) in both MCF7 and T47D [36] (Figure 5C). However, the B1SP–fulvestrant combination (Appendix A) exhibited an additive effect rather than synergistic. This would likely be due to differences in the mechanism of action of these two endocrine therapies.

### 3.6. Tamoxifen-Resistant ER^+^ Breast Cancer Remains Dependent on SEMA3C for Signaling and Growth

After exhibiting its functional and potential therapeutic role in tamoxifen-sensitive cells (MCF7, T47D), the next step was to understand the relevance of SEMA3C-dependent signaling in endocrine therapy resistant cells. To being with, we analyzed the expression levels of SEMA3C protein expression in tamoxifen-resistant (TamR) cells generated via two different conditions, representing distinct endocrine therapy strategies vs. tamoxifen sensitive (MCF7) ER^+^ breast cancer cell lines cells along with MCF10A [37]. We observed high expression levels of SEMA3C in tamoxifen resistant cells (TamC3 and TamR3) when compared with MCF7 (Figure 6A). This finding was consistent with the analysis of the global gene expression dataset from GSE67916, which compared the expression levels of genes between tamoxifen-sensitive (TamS) and -resistant cells (TamR) [23] (Figure 6B). Treating TamR cells with tamoxifen in their normal growth conditions over 48h had no effect or led to enhanced SEMA3C protein levels, suggesting the dependence of resistant cells on SEMA3C post-endocrine therapy resistance (Figure 6C). MCF7 cells had low expressions of SEMA3C under estradiol-deficient conditions, which mimicked tamoxifen’s mechanism of action, in which estradiol was unable able to activate the ER pathway (Figure 2C). Therefore, the next step was to explore whether these cells showcased any dependency on SEMA3C, as shown previously in tamoxifen-sensitive cells. It was clearly seen in our SEMA3C knockdown studies (25 nM) that it led to a decrease in the phosphorylation of known upregulated RTK pathways (EGFR) and ER expression, as observed in these cells and in the majority of tamoxifen resistant tumors [37] (Figure 6D), followed by a significant inhibitory effect on the growth of TamR and TamC3 cells (Figure 6E). As seen in (Figure 6F,G), these cells continued to show sensitivity to B1SP treatment for growth (1 uM) and signaling (0.5 uM). This might suggest, in some ER^+^ breast cancer cases, that the consistent activation of SEMA3C is exploited by the tumor cell population to evade treatment efficacy.

## 4. Discussion

Here, we examined the role of SEMA3C in the context of ER^+^ breast cancer, with the aim of defining SEMA3C as a novel potential therapeutic target for this breast cancer subtype. Notably, among the various subtypes of breast cancer, ER^+^ breast cancer exhibited the highest expression of SEMA3C mRNA in clinical samples. Importantly, we observed a positive association between the expression of SEMA3C and ER in patients with ER^+^ breast cancer. Our data further indicated that SEMA3C functions as an ER-induced growth factor that activates and maintains sustained proliferative and survival signaling pathways in both tamoxifen-sensitive and -resistant ER^+^ breast cancer cells. Significantly, SEMA3C pathway inhibition led to the attenuation of cell signaling, growth and ER expression of these cells. This study highlights the therapeutic potential of targeting SEMA3C in the treatment of ER^+^ breast cancer, with particular relevance for hormone-resistant cases.

Among all breast cancer subtypes, ER^+^ breast cancer expresses the highest levels of SEMA3C transcripts. This observation was consistent with the study presented by Zhang et al. [38]. Furthermore, our study revealed that SEMA3C protein levels were significantly higher in ER^+^ breast cancer cell lines MCF7 and T47D compared to the immortalized normal epithelial cell line MCF10A. These results are in line with the findings of Cole-Healy et al. [14], who reported that SEMA3C protein levels progressively increased during the transition from normal to invasive breast cancer.

This study demonstrates that SEMA3C is an ER-induced autocrine growth factor that drives the signaling and growth of ER^+^ breast cancer, mainly by activating MAPK and AKT pathways. Our findings validate previous research showing that SEMA3C knockdown inhibits the proliferation of ER^+^ breast cancer cells, and extends these insights by demonstrating that SEMA3C is an autocrine growth factor that stimulates the growth of ER+ cell lines. Importantly, our investigation revealed that SEMA3C activates MAPK and AKT pathways, which are essential for sustaining cancer cell proliferative signaling, a fundamental hallmark of cancer, making it a potential therapeutic target. SEMA3C also plays a critical role in the activation of phospho-EGFR levels, a key player in endocrine resistance of ER^+^ breast cancer cells [39], suggesting that targeting SEMA3C could be effective in overcoming endocrine resistance. 

We further identified ER as a key transcriptional regulator of SEMA3C expression in ER^+^ breast cancer cells, establishing SEMA3C as a tumor vulnerability in ER^+^ breast cancer and providing a mechanistic link between the SEMA3C-ER pathway and growth signaling in ER^+^ breast cancer. Targeting SEMA3C could be a potential strategy for combating ER^+^ breast cancer, highlighting its potential as a therapeutic target in precision oncology.

Importantly, we showed that SEMA3C is a novel, tractable therapeutic target for the treatment of ER+ BCa patients. Our findings revealed that B1SP treatment inhibited clinically relevant signaling pathways and the growth of ER^+^ breast cancer cells, similar to its effect on inhibiting prostate cancer growth [12]. These results suggest that targeting SEMA3C could be a possible therapeutic strategy for ER^+^ breast cancer patients and could provide a viable treatment option for patients who progress on endocrine therapy.

Combination therapies have emerged as a powerful approach for combating breast cancer [40], and our study suggests that targeting SEMA3C in combination with endocrine therapy could be a highly effective strategy. We have demonstrated the synergistic and additive effects of B1SP with tamoxifen and fulvestrant, the two most commonly used endocrine therapies for ER^+^ breast cancer. By selecting biologically relevant and non-lethal drug targets with the least side effects, we can ensure that patients receive the best possible treatment with minimal harm. Our study also highlighted the promising safety profile of SEMA3C inhibition, as SEMA3C-deficient mice are indistinguishable from their wild-type littermates, and Plexin-B1 deficient mice remain fertile and viable [41]. This provides further evidence that SEMA3C inhibition could be a well-tolerated and relatively safe option for breast cancer patients and should be further explored in clinical trials as a potential therapeutic approach.

There is a growing body of evidence linking EGFR, MAPK, and AKT signaling pathways to endocrine resistance in ER^+^ breast cancer [42,43]. Our study provides further support for this association, as we found that SEMA3C activates these pathways and drives tumor progression in tamoxifen-resistant cells. These findings suggest that targeting these pathways, either alone or in combination with endocrine therapy, could be a potential strategy for overcoming endocrine resistance and improving outcomes for ER^+^ breast cancer patients. 

A limitation of our study is the use of cell line models for ER^+^ breast cancer. While these models are widely used in breast cancer research and can provide valuable insights into cancer biology, they do not always fully recapitulate the complexity of the disease in vivo. Therefore, more research is needed in different model systems to further verify the importance of these findings. Additionally, further studies with larger sample sizes are needed to confirm our findings and to explore the potential of SEMA3C as a biomarker and therapeutic target for ER+ breast cancer.

## 5. Conclusions

Our findings on the role of SEMA3C in ER^+^ breast cancer have potential implications for breast cancer research and treatment. This study provided novel insights into the biology of ER^+^ breast cancer, highlighting the potential role of SEMA3C in driving tumor progression and maintaining tumor cell proliferation, especially in the context of endocrine resistance. These findings fit into the current understanding of breast cancer biology, which emphasizes the importance of targeting specific pathways and vulnerabilities in cancer cells for effective treatment. These results suggest that targeting SEMA3C could be a possible strategy for personalized medicine and precision oncology, especially for patients with ER^+^ breast cancer who are resistant to current therapies. Moreover, the study’s findings could pave the way for the development of new and more effective SEMA3C inhibitors with fewer off-target effects and less toxicity, leading to improved outcomes for breast cancer patients. Therefore, the study’s broader implications extend beyond the field of breast cancer to the broader context of cancer research and treatment.

## 6. Patents

A patent has already been approved for B1SP used in this paper (#WO 2010/111792 A1).

## Figures and Tables

**Figure 1 cells-12-01715-f001:**
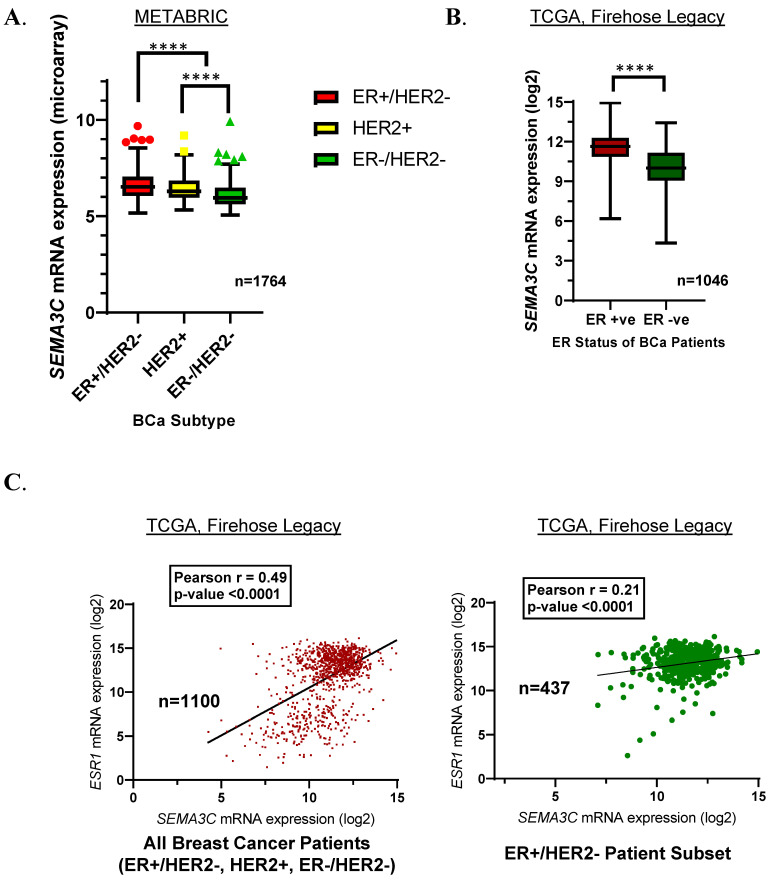
Semaphorin 3C (SEMA3C) mRNA expression associated with different breast cancer subtypes and correlation with estrogen receptor expression. (**A**) Analyzing a dataset of 1764 patients showed higher SEMA3C mRNA expression in the ER^+^/HER2^−^- breast cancer subtype, and (**B**) in the TCGA (Firehose Legacy) study we confirmed that SEMA3C expression was significantly higher in patients with ER^+^ diagnosis. (**C**) Furthermore, analyzing breast cancer patients (all subtypes or ER^+^/HER2^−^_ subset) in the TCGA study (Firehose Legacy) revealed ESR1 and SEMA3C mRNA expression had significant positive Pearson’s correlation (each dot represents data from one sample; **** *p* < 0.0001).

**Figure 2 cells-12-01715-f002:**
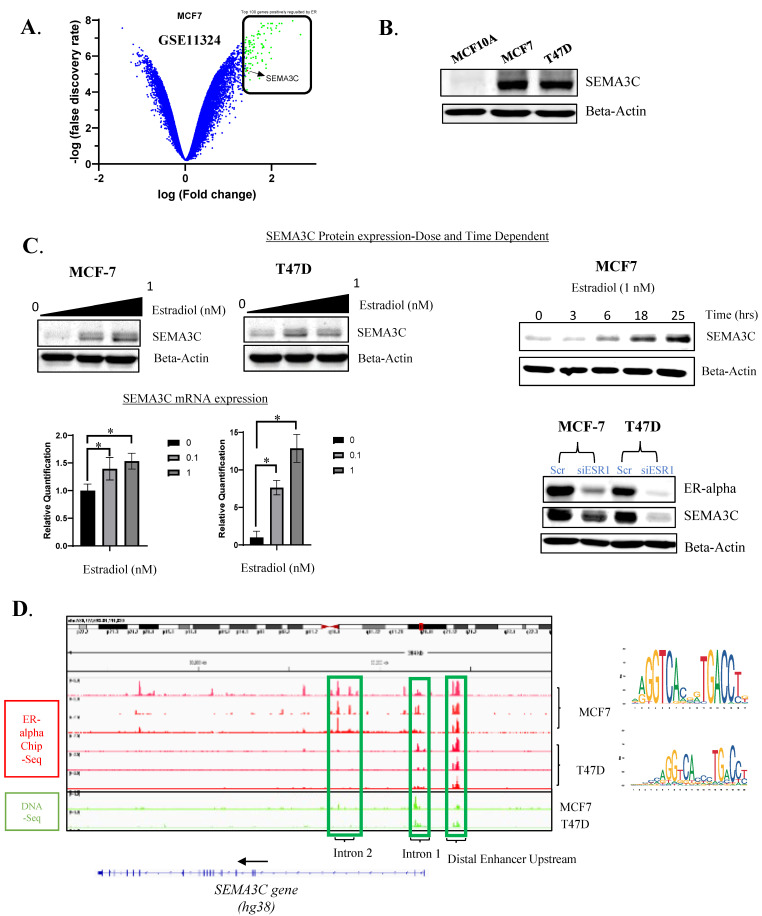
Semaphorin 3C (SEMA3C) expression confirmation in ER^+^ breast cancer cells and its regulation by ER in MCF7 and T47D. (**A**) Volcano plot analysis of differential gene expression dataset pre- and post-estrogen treatment (GSE11324). (**B**) Western Blot of SEMA3C protein expression in MCF10A, MCF7 and T47D. (**C**) Increase in protein and mRNA expression of SEMA3C by estradiol (E2) in a dose- and time-dependent manner along with siRNA ER-alpha knockdown, leading to a decrease in SEMA3C expression. (**D**) Confirmation of ER-alpha binding onto regulatory sites in the SEMA3C locus using ChIP-Atlas (IGV) and ERE binding motif sequences (MA0112.2, MA0112.3). * *p* < 0.05.

**Figure 3 cells-12-01715-f003:**
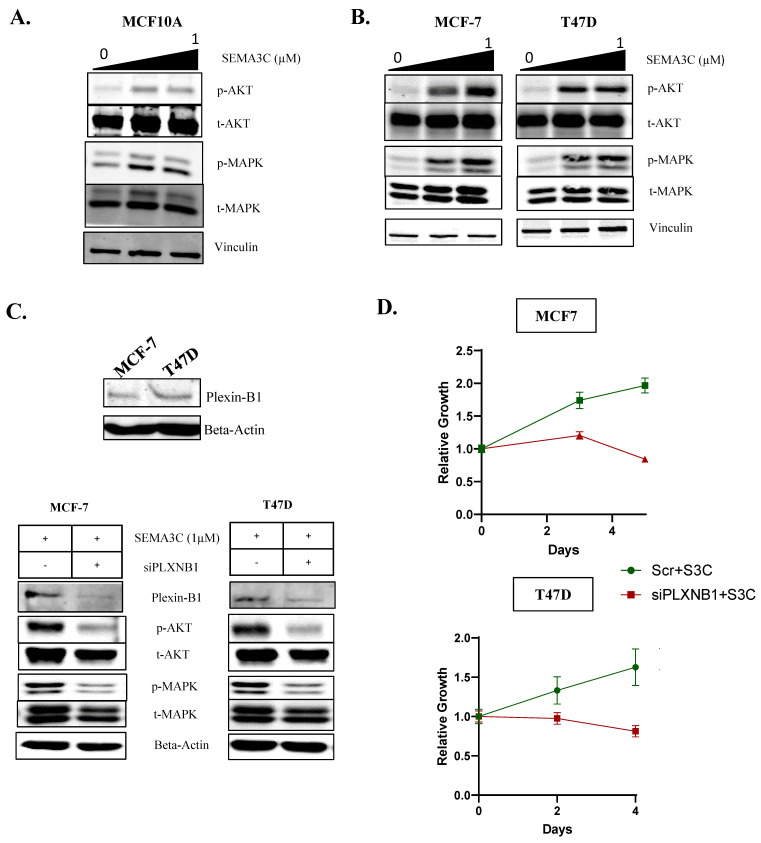
SEMA3C activates receptor tyrosine kinase pathways via the Plexin-B1 receptor in breast cancer cells. (**A**) Exogenous treatment with recombinant SEMA3C on MCF10A leads to an increase in phosphoactivation of AKT and MAPK. (**B**) Exogenous treatment with recombinant SEMA3C leads to the activation of AKT and ERK pathways in a dose-dependent manner in MCF7 and T47D. (**C**) Plexin-B1 siRNA knockdown (25 nM), showcasing the significant effect on SEMA3C signaling and (**D**)_impacting growth in MCF7 and T47D, confirming SEMA3C signal via Plexin-B1.

**Figure 4 cells-12-01715-f004:**
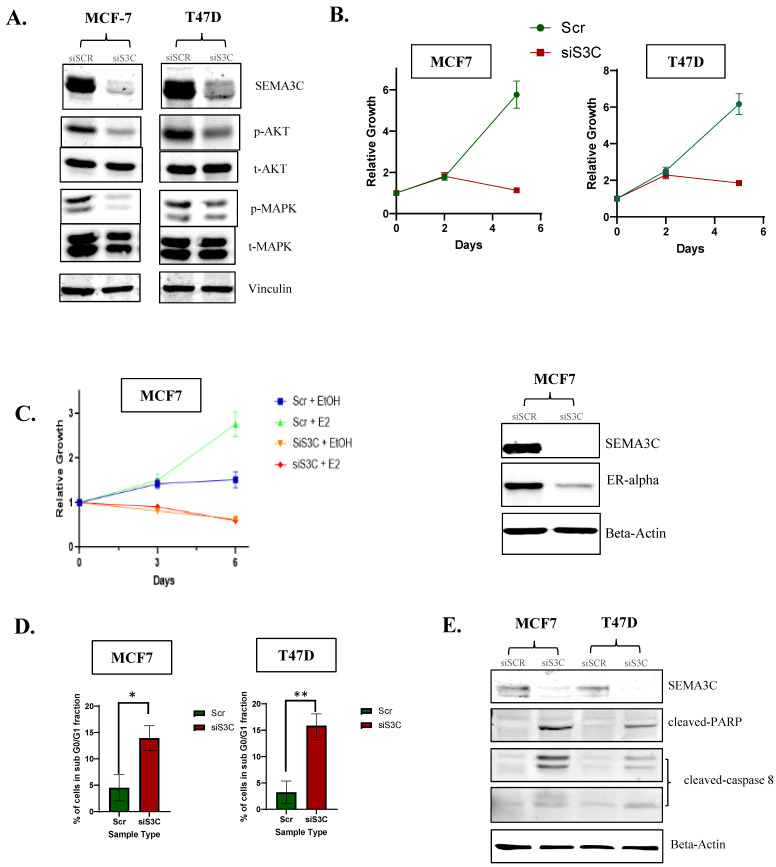
SEMA3C silencing in ER^+^ breast cancer cells leads to decreases in oncogenic signaling, cell survival and activation of pro-apoptotic pathways. (**A**) siRNA knockdown (25 nM) of SEMA3C in MCF7 and T47D cells resulted in downregulation of p-p44/42(Thr202/Tyr204)/Akt (Ser473) within 72hrs, and (**B**) led to cell growth inhibition in both of the cell models in vitro. (**C**) MCF7 showcased their dependency on SEMA3C, shown by the growth curve under different conditions (Scr or siSEMA3C/EtOH or Estradiol (1nM)), with an inability of estradiol to rescue growth, probably due to the downregulation of ER. (**D**) siRNA knockdown in MCF7 and T47D induced cell cycle arrest detected by flow cytometry with confirmation of (**E**) activation of the apoptotic pathway by an increase in expression of pro-apoptotic proteins c-PARP/c-Caspase-8 within 72 hrs. * *p* < 0.05; ** *p* < 0.01.

**Figure 5 cells-12-01715-f005:**
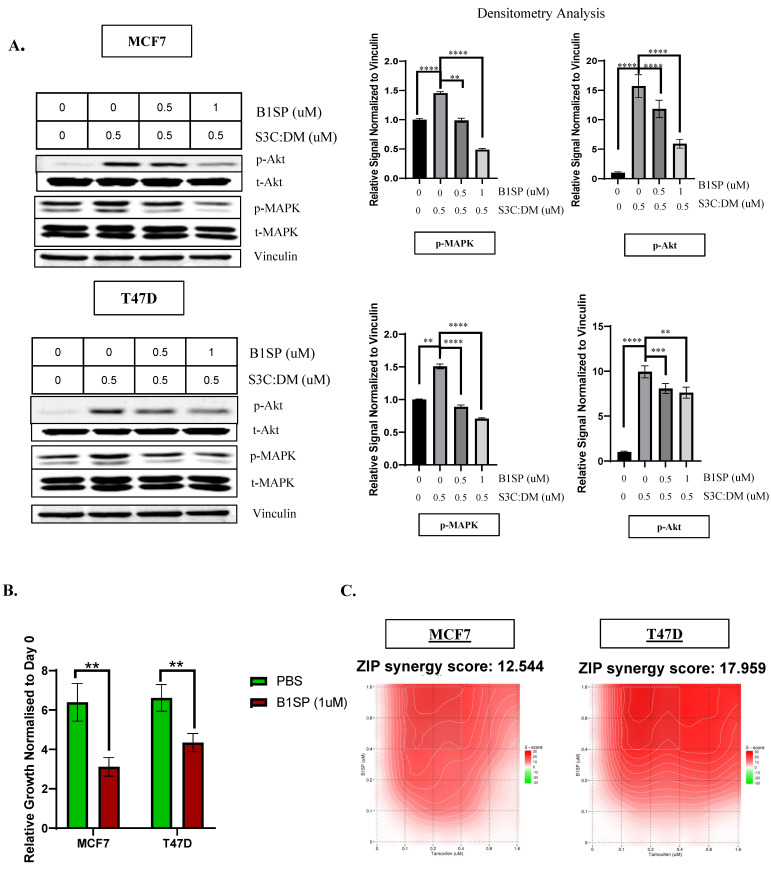
B1SP impacts SEMA3C: PlexinB1 mediated cell growth, signaling and showcasing positive combination effect with endocrine therapy drugs. (**A**) B1SP treatment in a dose-dependent manner affects SEMA3C-mediated signaling along with densitometry analysis. (**B**) B1SP-impacted cell growth within 72 h in the presence of full-serum conditions in MCF7 and T47D cells. (**C**) B1SP and tamoxifen treatment on tamoxifen-sensitive cells, showcasing synergistic behavior based on SynergyFinder Scoring. ** *p* < 0.01; *** *p* < 0.001; **** *p* < 0.0001.

**Figure 6 cells-12-01715-f006:**
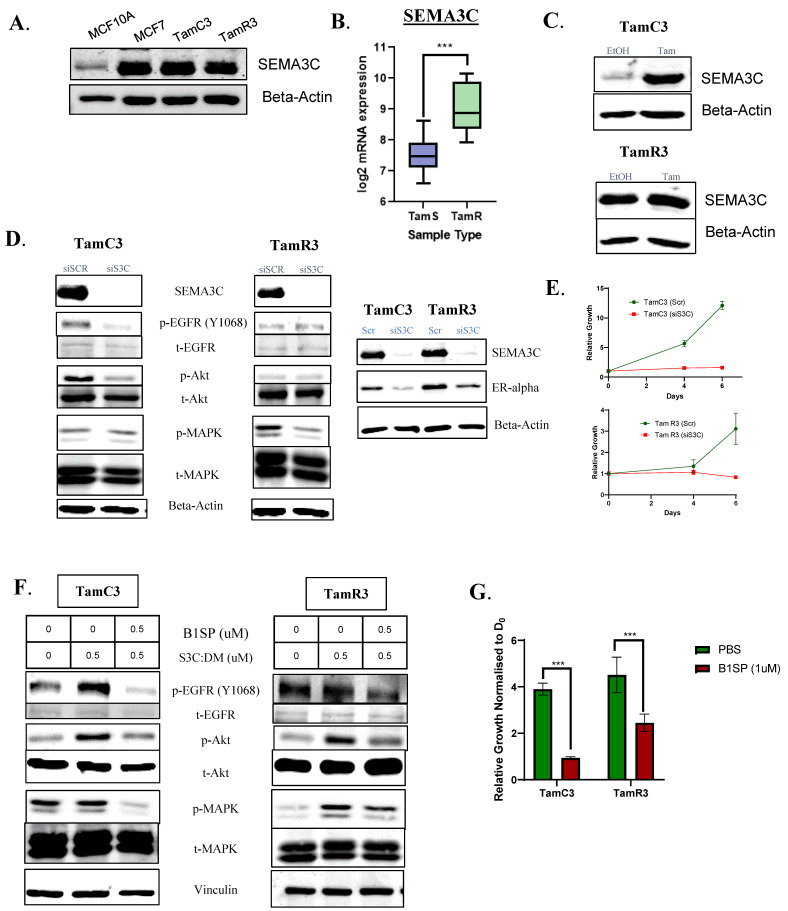
Tamoxifen-resistant cells continue to exhibit dependency on Semaphorin 3C (SEMA3C) for their growth and survival. (**A**) Basal-level SEMA3C protein expression in tamoxifen-sensitive (MCF7) vs. tamoxifen-resistant cells (TamC3, TamR3). (**B**) Global gene expression analysis of SEMA3C expression between MCF7-tamoxifen sensitive cells vs. MCF7-derived tamoxifen-resistant cells. (**C**) High levels of SEMA3C expression detected 48h post-tamoxifen treatment. SEMA3C siRNA silencing (**D**) downregulates receptor tyrosine kinase signaling (EGFR, Akt and MAPK) and estrogen receptor (ER) expression, (**E**) with growth inhibitory effect in TamR cells. (**F**,**G**) TamR cells respond to B1SP therapeutic effects, leading to significant decreases in SEMA3C-mediated cell signaling and growth. *** *p* < 0.001.

**Table 1 cells-12-01715-t001:** Detail of different studies utilised to analyze SEMA3C mRNA expression from studies containing tumor expression data from breast cancer patients and genome-wide differential expression cell-line based assays.

STUDY NAME	Number of Samples Utilized for Analysis	Database	Sample Grouping
Breast Cancer (METABRIC) [20,21]	1764	Cbioportal	ER^+^/HER2 ^−^ (n = 1257); HER2^+^ (n = 198); ER^−^/HER2 ^−^ (n = 309)
Breast Invasive Carcinoma (TCGA, Firehose Legacy) [22]	1101	Cbioportal	ER^+^/HER2 ^−^ (n = 437); All samples (n = 1101)
GSE67916 [23]	18	GEO	TamR1 (3 replicates); TamR4 (3 replicates); TamR7 (2 replicates); TamR8 (2 replicates); TamS (8 replicates)
GSE11324 [24]	12	GEO	0hr (3 replicates); 3hr (3 replicates); 6hr (3 replicates); 12hr (3 replicates)
GSE27473 [25]	6	GEO	MCF7-Scr (3 replicates); MCF7-silenced ER (3 replicates)
BRCA	1376	GEPIA2	Tumor (n = 1085); Normal (n = 291)

## Data Availability

Publicly available datasets analyzed in this study are mentioned in the Methods section of the paper.

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
