# Peer review of "Dependency of Tamoxifen Sensitive and Resistant ER+ Breast Cancer Cells on Semaphorin 3C (SEMA3C) for Growth"

_cells, 2023, doi:10.3390/cells12131715_

Round 1
Reviewer 1 Report
The manuscript entitled “Dependency of Tamoxifen Sensitive and Resistant ER+ Breast 2 Cancer Cells on Semaphorin 3C (SEMA3C) for growth” investigates the significance of SEMA3C in the progression of ER+ breast cancer and its association with resistance to endocrine therapy. The authors utilized in silico and in vitro approaches to validate the role of SEMA3C in ER signaling, growth of ER+ breast cancer and endocrine therapy resistance, suggesting its potential as a therapeutic target for patients with ER+ breast cancer. The manuscript is well designed and well structured and holds some potential for ER+ breast cancer treatment. However, I have the following comments and suggestions for the manuscript.
Abstract: The abstract is well written and results in abstract are described in a précised manner
Introduction is also well written and adequate background information is provided.
Methods: In section 2.3, I recommend that the authors provide more detailed information regarding each dataset. This should include the number of patients/subtypes included in each dataset, as well as the specific treatments or conditions used. Presenting this information in a tabular form can provide a better idea about the studies used in the manuscript.
Results: In Figure 1A, the authors examine the expression of SEMA3C in different breast cancer subtypes but do not include the ER+/HER2+ subtype. I suggest considering the inclusion of SEMA3C expression levels in the ER+/HER2+ subtype as well
For Figure 1C, where the authors explore the correlation between ESR1 and SEMA3C genes, it would be better to clarify whether the same patients were used as in the analysis of SEMA3C expression levels (Figure 1A/B). Additionally, I recommend examining the correlation between ESR1 and SEMA3C in ER+/HER2- breast cancer patients, as subsequent results are based on the co-expression of ESR1 and SEMA3C in this subtype.
Supplementary Figure S1 requires indications to distinguish between normal and tumor samples also authors should provide the significance levels/ p values etc.
Figure 2 A: Authors should also provide the dataset name/number on the top of volcano plot to avoid any confusion
Regarding Figure 2C, I suggest including the expression levels of ESR1 protein as a positive control following estradiol treatment in MCF7 and T47D cell lines.
In line 295-299 authors mentioned “ Next, to confirm whether SEMA3C is an ER-regulated gene, we treated MCF7 and T47D cells (previously grown in charcoal stripped serum for 4 days) with estradiol and monitored SEMA3C mRNA and protein levels. A significant dose dependent (0.1-1 nM) increase in SEMA3C mRNA and protein was observed after an overnight treatment with estradiol (Figure 2C)”. Wouldn’t it be more interesting if authors can see the protein expression levels of SEMA3C in time dependent manner as well?
For lines 335-336, where the authors treated MCF10A cells with recombinant SEMA3C, it would be beneficial to explain the rationale behind the chosen concentration. Additionally, clarify whether the observations were made after a 10-minute treatment with recombinant SEMA3C.
Figure 3 A-C: Authors should also consider providing the results of expression levels of SEMA3C protein post SEMA3C treatment along with AKT and MAPK signaling pathways. It would even be better to see the expression levels of ESR1 post SEMA3C treatment, this can further support the relationship/correlation between ESR1 and SEMA3C.
Figure 3 A : Did authors see any change in AKT or MAPK signaling post SEMA3C treatment in MCF10A cell line as well ?
There is a typographical error in Figure D
Figure 3 C: Some control are missing in these results, authors should consider including group without any treatment and a group with Just siPLXNB in both MCF-7 and T47D cell lines.
In line 379 : The authors should provide an initial explanation of the abbreviation "CSS" before using it in subsequent instances?
In line 388-389 authors have mentioned “ Through immunoblotting, we were able to confirm at protein levels an increase in cleaved caspase-3, and PARP (Figure 4E). MCF 7 cell line is reported to be caspase 3 deficient cell line and this has been reported by multiple studies (some of them are mentioned below). In cells that are deficient in caspase-3, the enzyme is either absent or not functioning properly. Therefore, caspase-3 deficient cells are unlikely to show detectable levels of cleaved caspase-3. I am just wondering how did the authors observe the expression levels of caspase 3 in MCF7 cell line???
1- Yang, Xiao-He, et al. "Reconstitution of caspase 3 sensitizes MCF-7 breast cancer cells to doxorubicin-and etoposide-induced apoptosis." Cancer research 61.1 (2001): 348-354.
2- Delom, F., et al. "Calnexin-dependent regulation of tunicamycin-induced apoptosis in breast carcinoma MCF-7 cells." Cell Death & Differentiation 14.3 (2007): 586-596.
The authors predominately used insilico approach and in vitro breast cancer cell lines as models to support majority of their conclusions, however, the xenograft based in vivo animal assay need to be included to further support the conclusion.
Reviewer 2 Report
positive correlation between BCa progression and SEMA3C overexpression exists, so authors to clearly align title and aim.
please note clear aim of the study was not provided.
justification of the selected methods was also not provided. instead, authors provided results analysis.
Provide version of Graphpad prism used
check for proper scientific writing e. 1x105 is not 100,000 cells
please do spell/grammar check
Reviewer 3 Report
The manuscript entitled “Dependency of Tamoxifen Sensitive and Resistant ER+ Breast 2 Cancer Cells on Semaphorin 3C (SEMA3C) for growth” and authored by Bhasin et al suggested that targeting SEMA3C could be a promising therapy especially for patients with ER+ breast cancer who are resistant to current treatments. Authors also argued that their presented findings may lead to the development of more effective SEMA3C inhibitors with fewer off-target effects and toxicity, leading to improved outcomes for breast cancer (and potentially other cancer) patients. One major concern is lacking proper control. For example, how normal breast cells relate to semaphorins?
Other comments
· Proofreading is required; One example, the “105” in first line of 2.6 Cell Stimulation should be revised 10 to the power of 5.
· In material and methods, quote “For B1SP production, we used protocol as described previously in [12].”, this method must be briefly included here.
· In figure 1 legend, explain what is the Firehose Legacy study?
· Include raw gels for all western blots as supplementary data.
· All western blots should be associated with proper quantification.
· References can use more updating.
proofreading is needed.
Round 2
Reviewer 1 Report
The authors have taken into account the majority of my comments and suggestions, and with some minor English language editing, the manuscript is now suitable for acceptance and publication.
Author Response
N/A
Reviewer 3 Report
Lack of control
ok
Author Response
MCF10A cells are commonly used as a control in many studies and are considered a widely accepted model for normal breast cells. While MCF10A cells are immortalized, they have been extensively characterized and possess many characteristics of normal breast cells, including their ability to form acini-like structures, express key markers of normal breast epithelial cells, and exhibit responses to various stimuli that are consistent with normal cellular behavior. We acknowledge the reviewer's concern and we agree that there are limitations associated with the use of immortalized cells as controls. We have added a sentence in the discussion: "Therefore, more research is needed in different model systems to further verify the importance of these findings."